# Isolation of a Novel Anti-Diabetic α-Glucosidase Oligo-Peptide Inhibitor from Fermented Rice Bran

**DOI:** 10.3390/foods12010183

**Published:** 2023-01-01

**Authors:** Jingfei Hu, Xiaohua Lai, Xudong Wu, Huanyu Wang, Nanhai Weng, Jing Lu, Mingsheng Lyu, Shujun Wang

**Affiliations:** 1Jiangsu Key Laboratory of Marine Bioresources and Environment/Jiangsu Key Laboratory of Marine Biotechnology, Jiangsu Ocean University, Lianyungang 222005, China; 2Co-Innovation Center of Jiangsu Marine Bio-Industry Technology, Jiangsu Ocean University, Lianyungang 222005, China

**Keywords:** bioactive oligo-peptides, α-glucosidase, rice bran, *Bacillus subtilis*, molecular docking

## Abstract

At present, the incidence rate of diabetes is increasing gradually, and inhibiting α-glucosidase is one of the effective methods used to control blood sugar. This study identified new peptides from rice bran fermentation broth and evaluated their inhibitory activity and mechanism against α-glucosidase. Rice bran was fermented with *Bacillus subtilis* MK15 and the polypeptides of <3 kDa were isolated by ultrafiltration and chromatographic column, and were then subjected to LC-MS/MS mass spectrometry analysis. The results revealed that the oligopeptide GLLGY showed the greatest inhibitory activity in vitro. Docking studies with GLLGY on human α-glucosidase (PDB ID 5NN8) suggested a binding energy of −7.1 kcal/mol. GLLGY acts as a non-competitive inhibitor and forms five hydrogen bonds with Asp282, Ser523, Asp616, and His674 of α-glucosidase. Moreover, it retained its inhibitory activity even in a simulated digestion environment in vitro. The oligopeptide GLLGY could be developed into a potential anti-diabetic agent.

## 1. Introduction

Diabetes is a complex multifactorial disease. Type 2 diabetes mellitus (T2DM) has become a serious health concern in developing countries [1], and is caused by insulin resistance due to altered β-insulin secretion or deficiency [2]. Worldwide, the incidence rate of diabetes is increasing rapidly, including in China. The development of diabetes is a very long process which is often accompanied by other complications such as diabetic foot, diabetic nephropathy, and cerebral thrombosis [2,3]. Delaying glucose absorption by inhibiting the activity of carbohydrate digestive enzymes such as α-glucosidase is one of the effective ways to control blood glucose levels [4]. To date, acarbose, voglibose, and miglitol are the three main α-glucosidase inhibitors used in the clinic [4]. These show good inhibitory effects but have certain side effects. Also, because of their long synthetic pathway it is necessary to find other α-glucosidase inhibitors with acceptable side effects. Natural products are rich in peptides, polysaccharides, phenolic compounds, terpenoids, and other bioactive substances that have a variety of functional properties [5,6]. The bioactive peptides extracted from *Spirulina platensis* [7], oat protein [8], canary seeds [9], quinoa [10], and other natural substances have inhibitory effects on α-glucosidase. In recent years, bioactive peptides among bioactive compounds have become the most important compounds for the health and food industry, and can be used as substitutes for drug therapy to control lifestyle-related diseases [11]. Therefore, the inhibition of α-glucosidase by bioactive peptides is a good candidate for further research.

China, a large producer of rice, produces ample rice bran from rice processing. However, rice bran utilization is low, as it is mainly used as feed. This resource is being seriously overlooked [12]. Rice bran contains various bioactive substances that are beneficial to human health, such as amino acids, phenolic acids, flavonoids, vitamin E, γ- oryzanol, and pigment compounds. *Bacillus subtilis,* a popular biological starter, can produce protease, lipase, and cellulase during its growth process. It is often used for the hydrolysis of beans to produce amino acids, peptides, and amines [13]. Microorganisms can make complete use of rice bran nutrients for their growth and biochemical metabolism. They can metabolically modify, recombine, transfer, or rearrange active groups to improve the functional and processing characteristics of rice bran peptides [14]. The rice bran fermentation broth contains high-quality proteins [15]. The bioactive peptides produced from rice bran proteins are easy to absorb and have non-toxic side effects. Studies suggest that these bioactive peptides have inhibitory qualities and work against cancer, α-glucosidase, β-glucosidase, α-amylase, and angiotensin-converting enzymes [16]. Therefore, they are widely used in the food and pharmaceutical industry and have become a popular research topic [17].

Molecular docking has been used for the development of health products and drug research. In vitro and in vivo tests for ligand screening are quite time-consuming and expensive [18,19,20]. On the other hand, molecular docking can help screen ligands more quickly. This method has been extensively used in the research of health products and drugs [21,22]. Molecular docking can be of three types: rigid docking, semi-flexible docking, and flexible docking. Most docking programs are based on the semi-flexible type of docking. In this method, the macromolecular protein is fixed, and the conformations of the small molecule ligand are changed during the docking process to simulate the ligand docking on protein [23]. The docking predictions made possible by molecular docking can greatly shorten research time, improving research efficiency.

This study aimed to isolate bioactive peptides from the fermentation rice bran that inhibits α-glucosidase. The potential peptides that inhibited α-glucosidase were further purified by ultrafiltration and chromatography. Next, the peptides were identified by LC-MS/MS. Furthermore, molecular docking simulation studies were performed and enzyme inhibition mechanisms on α-glucosidase were investigated. Finally, the best oligopeptide obtained from the fermented rice bran was tested in a simulated GI digestion system in vitro. Our results may improve the economic benefits of rice bran.

## 2. Materials and Methods

### 2.1. Materials

Rice bran was provided by Yancheng Kanglinda Biotechnology Co., Ltd. (Yancheng, China). *Bacillus subtilis* MK15 was provided by the Provincial Institute of Marine Resources Development (Lianyungang, China) and cultured in Luria–Bertani (LB) medium. α-glucosidase, α-amylase, porcine pepsin, pancreatin, bile extract porcine, and 4-Nitrophenyl α-D-glucopyranoside (pNPG) were purchased from Sigma Co., St. Louis, Missouri, USA. Fish meal peptone, glucose, NaCl, urea, beef extract, sodium carbonate, Na_2_HPO_4_·12H_2_0, NaH_2_PO_4_·2H_2_0, absolute ethanol, and trichloroacetic acid (TCA) were purchased from Aladdin, Shanghai, China. DEAE Sepharose fast flow ion exchange resin and Sephadex G-25 were acquired from Auyoo Biotechnology Co., Ltd. (Shanghai, China).

### 2.2. Preparation of Seed and Fermentation Broth

*B. subtilis* MK15 was inoculated into a 100 mL seed culture medium (NA: beef extract 0.3%, peptone 1%, NaCl 0.5%, pH 7.0, and tap water; sterilized at 121 °C for 20 min), and then incubated in a shaking table at 180 rpm and 37 °C for 12 h [24]. The 60 mL of seed culture was inoculated (3%) into 2 L of rice bran fermentation medium (rice bran 5%, glucose 0.1%, NaCl 0.5%, urea 0.5%, pH 8.0, and tap water; sterilized at 115 °C for 20 min), and then incubated in a shaking table at 35 °C and 180 rpm for 48 h. The fermented broth was centrifuged at 8000 rpm and 4 °C for 30 min to collect the supernatant [25]. The initial volume of fermentation medium was 2 L, and the supernatant obtained was 1.82 L.

### 2.3. Fermentation Product Extraction

The supernatant was added to four times the amount of absolute ethanol for precipitation at 4 °C for 24 h and the mixture was centrifuged at 8000 rpm for 30 min. Next, the obtained supernatant was evaporated in a rotary evaporator at 45 °C to reduce the mixture volume and paused every 10 min (to prevent protein denaturation). This was repeated approximately 10 times. 1.82 L supernatant, evaporation time was 2 h, and the remaining volume of the final supernatant was 150 mL. Subsequently, an equal volume of 10% TCA was added and the mixture was rotated at 4 °C for 1 h with a magnetic stirrer. After centrifugation at 8000 rpm, the obtained supernatant was subjected to rotary evaporation until there was no flowing liquid. Next, 50 mL of water was added for re-dissolution. Finally, after centrifugation, the obtained precipitate was freeze-dried (FreeZon^®^2.5 L, Labconco, Kansas City, MO, USA) [26].

### 2.4. Determination of α-Glucosidase Inhibitory Activity

The activity assay was improved based on an in vitro assay model [27]. 50 μL (0.5 mol/L) pH 6.7 phosphate solution, 30 μL (0.5 mM) pNPG substrate solution, 50 μL rice bran fermentation extract, and 20 μL α-glucosidase solution (0.3 U/mL in phosphate solution) were mixed in a 96-well plate. The mixture was incubated at 37 °C for 30 min and then added to a 50 μL (0.67 mol/L) Na_2_CO_3_ solution to terminate the reaction. The sample absorbance was measured at 405 nm (Multiskan GO, Thermo Scientific, Waltham, MA, USA). Acarbose was used as a positive control. The blank group used phosphate solution instead of pNPG substrate and enzyme solutions; the control group used phosphate solution instead of the sample solution. The α-glucosidase inhibition rate of fermentation products was calculated as follows:α-glucosidase-inhibitory activity (%) = [1 − ((A_a_ − A_b_)/A_c_)] × 100(1)
where: A_a_, A_b_, and A_c_ are the absorbance values of the sample, blank, and control groups, respectively.

### 2.5. Separation and Purification

#### 2.5.1. Ultrafiltration

The concentration of crude rice bran peptide solution was 30 mg/mL, which was separated by ultrafiltration membranes with a molecular weight cutoff of 10 and 3 kDa in a serial manner to collect various components such as >10 kDa, 3–10 kDa, and <3 kDa. The three fractions were prepared with the same concentration (50 mg/mL) to determine their α-glucosidase inhibitory activity [28].

#### 2.5.2. DEAE Sepharose Fast Flow Ion Column Chromatography

10 mL (50 mg/mL) of bioactive peptide with molecular weight less than 3 kDa was filtered with a 0.22 μM membrane before separation by DEAE Sepharose fast flow ion column chromatography (column size: 20 cm × 4 cm, one-time elution volume was ~230 mL). The column was successively passed through with distilled water, 0.1 M NaCl, 0.2 M NaCl, 0.3 M NaCl, 0.7 M NaCl, and 1 M NaCl at a flow rate of 1 mL/min. Different collected fractions were detected at 280 nm and lyophilized after elution [28,29].

#### 2.5.3. Sephadex G-25 Column Chromatography

As seen earlier, after filtration with 0.22 μm membrane, 10 mL (50 mg/mL) of the sample purified by the ion column was passed through a Sephadex G-25 column (column size: 100 cm × 1.6 cm, one-time elution volume was ~190 mL) and then eluted. The flow rate was 1 mL/min, and the components were monitored at 280 nm and lyophilized after elution [28,29].

### 2.6. Identification of Rice Bran Peptides

The respective samples were sent to the DG peptides company Dgpeptides Co., Ltd., Hang Zhou, China for LC-MS/MS mass spectrometry analysis.

Sample Preparation: after reduction by 10 mM DL-dithiothreitol (DTT) at 56 °C for 1 h and alkylation by 50 mM iodoacetamide (IAA) at room temperature for 40 min in the dark, the extracted peptides were lyophilized. The peptides were re-suspended in 2–20 μL of 0.1% formic acid for LC-MS/MS analysis.

Nano-LC details were as follows: Nanoflow UPLC: Easy-nLC 1200 (Thermo Fisher Scientific, MA, USA); Nanocolumn: 150 μm × 15 cm in-house-made column packed with a reversed-phase ReproSil-Pur C18-AQ resin (1.9 μm, 100 Å, Dr. Maisch GmbH, Germany); loaded sample volume: 5 μL; mobile phase: A: 0.1% formic acid in water; B: 20% 0.1% formic acid in water −80% acetonitrile; total flow rate: 600 nL/min; LC linear gradient: 6 to 9% B for 5 min, 9 to 14% B for 15 min, 14 to 30% B for 30 min, 30 to 40% B for 8 min, and 40 to 95% B for 2 min.

Select up to 20 strongest peptide ions from the preview scan of Orbitrap. Data analysis: The raw MS files were analyzed and searched against the protein database based on the species of the samples using Byonic. The parameters were set as follows: the protein modifications were carbamidomethylation (C) (fixed), oxidation (M) (variable); the enzyme specificity was set to non-specific; the maximum missed cleavages were set to 3; the precursor ion mass tolerance was set to 20 ppm, and MS/MS tolerance was 0.02 Da. Only the peptides with high confidence were selected for downstream protein identification analysis.

### 2.7. Molecular Docking Studies

Docking with Autodock Vina 1.1.2.The 3D X-ray crystal structure of the N-terminal domain of human lysosomal α-glucosidase (PDB ID 5NN8; resolution 2.45 Å) was retrieved in PDB format from the protein database (www.rcsb.org, accessed on 6 March 2022). Subsequently, the cocrystallized ligands and water molecules were removed from the protein structure using PyMOL version 2.5.0. The 2D structure of the polypeptide was drawn by KingDraw software and converted into a 3D structure in Chem3D 20.0 software. Jiang’s butt joint size method [30] was used with little improvement. The search grid of the α-glucosidase was identified as center x −13.389, center y −38.216, and center z 95.021 with dimension sizes x 27.0, y 23.25, and z 23.25, spacing (angstrom) 0.375. The value of exhaustiveness was set to 9. The other parameters were set to default values. The binding energy of each active peptide was recorded. The hydrogen bond interactions between the receptor and ligand were analyzed by PyMOL version 2.5.0.

### 2.8. α-Glucosidase Inhibition by Synthetic Peptides In Vitro

The peptides with good binding ability in molecular docking studies were synthesized and tested for in vitro α-glucosidase inhibitory activity as in Section 2.4. The peptides were purchased from DGpeptides Co., Ltd. (Hangzhou City, Zhejiang Province, China) and purified to >98%. Their identity was confirmed by LC-MS/MS.

### 2.9. Mechanism of α-Glucosidase Inhibition

The inhibition kinetics of the oligopeptide GLLGY were tested to determine its efficiency. The sample GLLGY concentration was 1 mg/mL and the substrate pNPG concentrations were 0.482, 0.241, 0.121, and 0.060 mM. The Lineweaver–Burke diagram plots were constructed by initial velocity data to determine the enzyme Km (Michaelis constant), Vmax (maximum velocity), and Ki (inhibition binding constant) [24].

### 2.10. In Vitro Simulated Gastrointestinal (GI) Digestion System

According to the experimental method of Minekus et al. [31], 5 mL of sample (1 mg/mL) was mixed with 3.5 mL of simulated salivary fluid (SSF), 0.5 mL of α-amylase prepared by SSF (1500 U/mL), 25 μL of CaCl_2_ (0.3 M), and 975 μL of ionic water, and reacted at pH 7 and 37 °C for 2 min. 1 M HCl regulated the orally digested sample to pH 3.0 to terminate the oral digestion. 3.75 mL simulated gastric juice (SGF) was added for mixing, as well as 0.8 mL porcine pepsin prepared by SGF (25,000 U/mL), 2.5 L CaCl_2_ (0.3 M), and 0.348 μL ionized water, reacted at 37 °C for 2 h. 1 M NaOH adjusted the stomach digested sample to pH 7.0 to terminate the gastric digestion experiment, 2.75 mL simulated intestinal fluid (SIF) was added for mixing, as well as 1.25 mL of pancreatin prepared by SIF (800 U/mL), 0.625 mL of bile extract porcine (160 mM), 10 μL of CaCl_2_ (0.3 M), and 0.328 mL of deionized water, reacted at 37 °C for 2 h. The sample was boiled in water for 5 min, and centrifuged to collect the supernatant.

### 2.11. Statistical Analysis

All experiments were conducted in three parallel groups. Data are presented as means ± SD and analyzed (SPSS 17.0). The threshold for statistical significance was *p* < 0.05.

## 3. Results

### 3.1. Isolation, Purification, and Identification of Rice Bran Peptides

#### 3.1.1. Inhibition Rate of Fermentation Products against α-Glucosidase

The crude peptides extracted from the rice bran fermentation broth showed a lower inhibitory effect on α-glucosidase compared with acarbose (Figure 1a); the inhibition rate increased slowly at 20–50 mg/mL and rapidly at 50–80 mg/mL. This is because a higher polypeptide concentration improved binding to the substrate pNPG. The inhibition rate became stable at 80–100 mg/mL, suggesting saturated binding between the substrate and polypeptide. A further increase in the concentration showed no improvement in the inhibitory effect [32].

#### 3.1.2. α-Glucosidase Inhibitory Activity of Ultrafiltered Rice Bran Peptides

As shown in Figure 1b, peptides in the ultrafiltration component <3 kDa showed the highest inhibition rate (86%), and peptides of the 3–10 kDa component showed the lowest inhibition rate (44%). The inhibition rate of peptides >10 kDa was 50%. These results are consistent with Ibrahim et al. They showed that the inhibitory peptides of α-glucosidase were <2 kDa [33]. Therefore, peptides <3 kDa were selected for further analysis.

The inhibition rate of ultrafiltration component <3 kDa was determined at different concentrations (Figure 1c); the inhibition rate increased significantly at 45–60 mg/mL. This showed the inhibition of α-glucosidase is a concentration-dependent phenomenon.

#### 3.1.3. Purification by DEAE Sepharose Fast Flow Ion Column

The purification chromatogram of the polypeptide by DEAE Sepharose fast flow ion column is shown in Figure 2a. Two peaks were detected at 280 nm. Peak 1 components RB_1_ (neutral peptides; Xu et al., 2020) were eluted (peak fractions) with ultrapure water; peak 2 components RB_2_ (negatively charged) were eluted (peak fractions) with 0.1 M NaCl. Notably, the α-glucosidase inhibitory effect of RB_1_ was higher than that of RB_2_. Accordingly, RB_1_ components were further purified (Figure 2a).

#### 3.1.4. Purification by Sephadex G-25 Column and Identification

The RB1 components were eluted with pure water through a Sephadex G-25 chromatographic column, and 5 peaks were observed in the chromatogram (Figure 2b). The α-glucosidase inhibitory effect of 4 peaks was measured, while peak 5 was not tested due to very small amounts of testable material available. The inhibitory effect of the peak 4 components was the best-performing at 40% (Figure 2b), and was lyophilized for peptide component identification by LC-MS/MS mass spectrometry.

### 3.2. Identification and Molecular Docking of Rice Bran Peptides

The LC-MS/MS of the rice bran peptide is shown in Appendix A. The identified peptides were docked with Autodock Vina 1.1.2 software to understand their binding ability and mechanism with α-glucosidase. The molecular docking results are shown in Table 1. The docking binding energy of acarbose is −7.4 kcal/mol, with 11 hydrogen bond interactions with Asp282, Asp281, Asp404, Arg600, Asp616, and His674 of α-glucosidase. Among the identified peptides, LFSGF, HWP, QSFF, FPF, FSGF, VYGF, GLIGY, and GLLGY showed binding energy close to that of acarbose (>−7.0 kcal/mol). These peptides have <6 amino acid residues [34]. This suggests that short-chain peptides have a better inhibitory effect against α-glucosidase [34]. Oligopeptide PSR had three amino acids and 10 hydrogen bonds, which were synthesized for later experiments. Oligopeptide KGDPY and polypeptides QSFLQRYYFLFRILPL, QSFLQRYYFLFRILPL, ISIFLSFFFGLIAGT were not synthesized for the next experiment due to their low docking score.

### 3.3. In Vitro α-Glucosidase Inhibition Activity

Several peptides with excellent docking results were selected for an in vitro inhibition test against α-glucosidase. As shown in Figure 3a, GLLGY had the best inhibition effect (15%) at the equivalent concentration of 1 mg/mL. The α-glucosidase inhibition rates of FPF, VYGF, and LFSGF were <6%, and the inhibition rate of GLIGY and PSR were <3%. The inhibitory effect of the GLLGY is modest. Meanwhile, the inhibition rate against α-glucosidase increased with the increase in the concentration of GLLGY (Figure 3b). The LC-MS/MS of the GLLGY sample is shown in Appendix A and the most abundant peptide (M/Z 522.292) is shown in Appendix A.

### 3.4. Characteristics of Oligopeptide GLLGY

The features of GLLGY were as follows: isoelectric point, 6.2; hydrophilic residues, 0%; average hydrophilicity, −1.1; net charge, 0.0 at pH = 7.0 (https://pepcalc.com/ accessed on 10 May 2022); the solubility of oligopeptide GLLGY in pH 3 and pH 7, 100%. Figure 4a depicts the 3D structure of α-glucosidase. Figure 4b shows the structure of oligopeptide GLLGY. The interactions between oligopeptide GLLGY and α-glucosidase were simulated by molecular docking. Oligopeptide GLLGY binds to the active site and has five hydrogen bond interactions with α-glucosidase: Asp282, 3.1 Å; Ser523, 2.3 Å; Asp616, 2.2 Å and 3.2 Å; His674, 2.9 Å (Figure 4c, Table 1). The binding energy of GLLGY is −7.1 kcal/mol.

### 3.5. α-Glucosidase Inhibitory Mechanism of Oligopeptide GLLGY

Figure 5 showed that GLLGY is a non-competitive inhibitor of α-glucosidase. Vmax decreased and Km remained unchanged. For α-glucosidase, the calculated Ki of GLLGY is 0.724 mM pNPG. GLLGY is linked to the α-glucosidase binding sites, which reduces the activity of α-glucosidase [35]. The increase in substrate concentration does not reduce the inhibitory effect. Non-competitive inhibitors usually have reversible binding properties and do not affect the substrate binding process as they can act on various sites, inducing conformational changes affecting enzyme activity [9].

### 3.6. Effect of Simulated Gastrointestinal (GI) Digestion on the α-Glucosidase Activity of GLLGY

The results of in vitro GI simulation digestion are shown in Figure 6. After gastric and gastrointestinal digestion, the oligopeptide GLLGY still remained active, showing 83.37% inhibition rates against α-glucosidase. Statistically, though the α-glucosidase inhibitory effect of GLLGY decreased significantly, it retained strong anti-digestion ability in both the stomach and intestine.

## 4. Discussion

The extraction of active substances by microbial fermentation is emerging as a promising method [36]. Natural antidiabetic drugs have been extracted from microbial fermented food substances [37]. Rice bran is rich in proteins [5]. Like the α-glucosidase inhibitor peptides obtained from fermented rubing cheese [38], Chinese pickled chili peppers [39], and soybean condiments [40], the bioactive peptides extracted from rice bran fermentation broth can inhibit α-glucosidase. Bioactive peptides inhibit α-glucosidase by inhibiting the competition between oligosaccharides and α-glucosidase activity in small intestinal parietal cells, which slows down the intestinal absorption of glucose, reducing blood glucose levels [41]. The short-chain peptides not only maintain their activity during gastrointestinal digestion and hydrolysis, but they are also suitable for design and chemical synthesis [33]. Several studies showed that peptides resistant to GI digestion have lower hydrophobicity, and more positive net charge at pH 7.0. [42,43]. The average chain length and molecular weight of stable peptides are 4.5 ± 2.0 amino acid residues and 547.78 ± 233.17 g/mol, respectively, with a slightly positive net charge. Peptides with lower molecular weight may possess fewer protease recognition and cleavage sites [43]. The oligopeptide GLLGY is consistent with the peptide stability requirements of the in vitro GI digestion system. Moreover, the oligopeptide GLLGY maintained a good inhibitory effect on α-glucosidase in the simulated GI digestion system in vitro.

After a series of purifications, we found that the bioactive peptides were mainly <3 kDa. Thus, directly purifying components <3 kDa can save a significant amount of time. This approach is preferably applicable to the anti-diabetic peptides derived by fermentation from rice bran. A higher molecular weight peptide can have steric hindrance while binding to α-glucosidase [34]. The docking results of the purified peptide and alpha glucosidase molecule by ion exchange and Sephadex gel chromatography (Figure 2) proved that the peptide with lower molecular weight has a better inhibitory effect. GLLGY forms five hydrogen bonds with Asp282, Ser523, Asp616, and His674 residues of α-glucosidase (Figure 4c). It binds into the active site of α-glucosidase to prevent the entry of substrate-altering enzyme conformation that is no longer suitable for activity [44]. Jiang et al. [30] reported four hydrogen bonds between GSR and Asp282, Asp518, and Asp616 of α-glucosidase (PDB ID 5NN8), and five hydrogen bonds between EAK and His674, Asp518, Arg600, Asp616, and Asp282 of α-glucosidase. GLLGY binds to Asp282, Ser523, Asp616, and His674 residues to produce five hydrogen bonds. This shows that even with different amino acids, GLLGY binds to α-glucosidase like oligopeptide Gly-Ser-Arg and Glu-Ala-Lys.

In vitro activity assays showed that GLLGY inhibited the activity of α-glucosidase more than the other four peptides. The inhibition rate of GLLGY at 1 mg/mL was 15%, which is lower than that of hemp seed protein PLMLP (IC_50_ at 0.024 mg/mL) [45]. However, the α-glucosidase inhibition rate of GLLGY (1 mg/mL) is comparable to HNKPEVEVR (1.25 mg/mL), found in soft-shelled turtle eggs [34]. Notably, acarbose is a well-known competitive inhibitor of α-glucosidase. On the other hand, we found that GLLGY is a non-competitive inhibitor. Studies showed that the active site of α-glucosidase is mainly formed by acidic (Asp518, Glu521, and Asp616) and basic (Arg600 and His674) residues [46,47]. Asp518 acts as a nucleophile, while Glu521 and Asp616 stabilize the transition state through acid–base catalysis with Arg600 and His674. Past studies have shown that molecular docking is a reliable method to screen inhibitors of α-glucosidase [4]. In molecular docking studies, acarbose binds to Arg600, Asp616, and His674 residues, while GLLGY binds to Asp616 and His674 residues. According to the Lineweaver–Burk plots, the GLLGY is a non-competitive inhibitor. Compared with Acarbose, GLLGY has a different hydrogen bond distance with Asp616 and His674. Based on the structure of α-glucosidase, Asp616 and His674 are the key amino acid residues of the catalyst domain. They might be set as targets of molecule docking of α-glucosidase inhibititors. Leucine was considered important for enhancing insulin secretion through metabolic variable structure activation or membrane depolarization [45,48]. GLLGY binds to α-glucosidase via five hydrogen bonds, which may be the reason for its higher inhibition rate compared to other oligopeptides. Overall, GLLGY is a novel anti-diabetic peptide and the exact mechanism of its inhibition of α-glucosidase activity remains unclear and should be explored further.

## 5. Conclusions

Rice bran was fermented by *Bacillus subtilis* MK15 to extract the peptide components. The rice bran peptides were identified by LC-MS/MS. Oligopeptide GLLGY showed the best inhibitory effect against α-glucosidase. Molecular docking results suggested that it forms 5 hydrogen bonds and van der Waals interactions involving the second Leu (GLLGY) with α-glucosidase. GLLGY is a non-competitive inhibitor of α-glucosidase. In addition, GLLGY maintained an excellent α-glucosidase inhibitory effect in a GI digestion system. GLLGY is a promising anti-diabetic candidate and must be researched further. So far, the inhibitory effect was only studied in vitro; the effect in vivo still needs further exploration.

## Figures and Tables

**Figure 1 foods-12-00183-f001:**
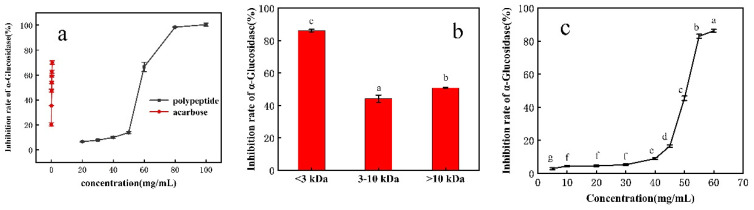
α-glucosidase inhibition activity of (**a**) crude rice bran peptides, (**b**) bioactive peptides after ultrafiltration, and (**c**) bioactive peptides of <3 kDa.

**Figure 2 foods-12-00183-f002:**
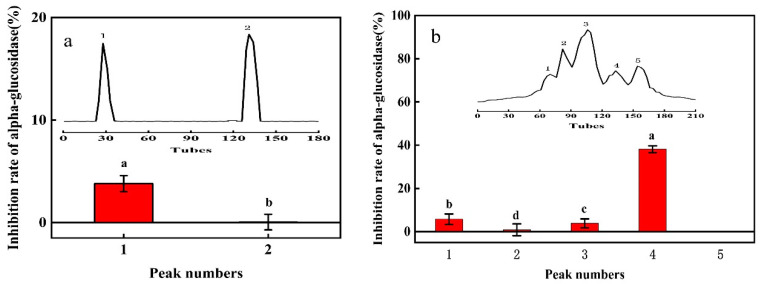
(**a**) Chromatogram (insert) and α-glucosidase inhibitory activity of bioactive peptides purified with a DEAE Sepharose fast flow ion column. (**b**) Chromatogram (insert) and α-glucosidase inhibitory activity of bioactive peptides purified with a Sephadex G-25 column.

**Figure 3 foods-12-00183-f003:**
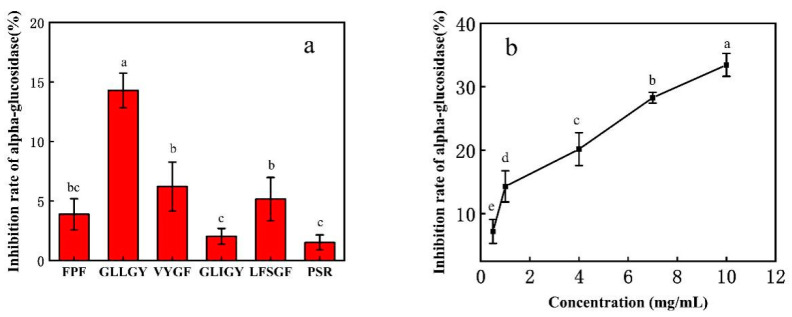
The α-glucosidase inhibitory activity of (**a**) peptides and (**b**) oligopeptide GLLGY.

**Figure 4 foods-12-00183-f004:**
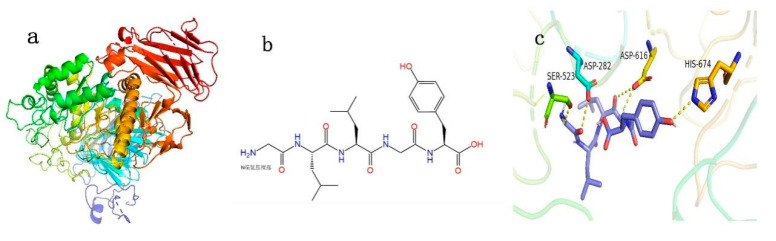
The binding interactions between oligopeptide GLLGY and human α-glucosidase. (**a**) Three-dimensional structure of α-glucosidase. (**b**) The structure of oligopeptide GLLGY. (**c**) Interaction diagram of GLLGY with α-glucosidase. The yellow dotted lines indicate hydrogen bonds.

**Figure 5 foods-12-00183-f005:**
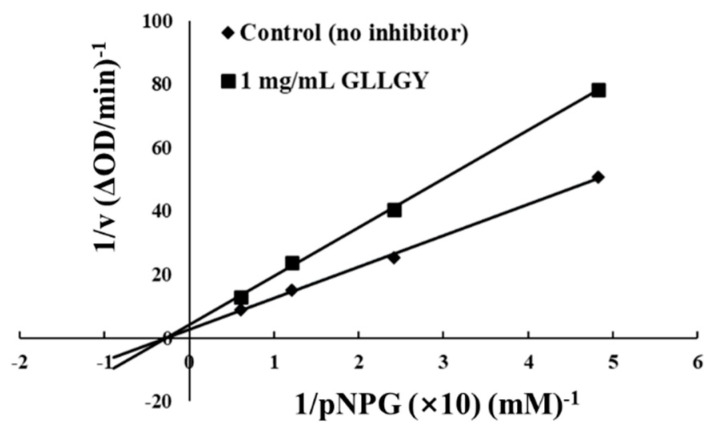
Lineweaver−Burk plots of α-glucosidase catalyzed reactions in the presence of GLLGY.

**Figure 6 foods-12-00183-f006:**
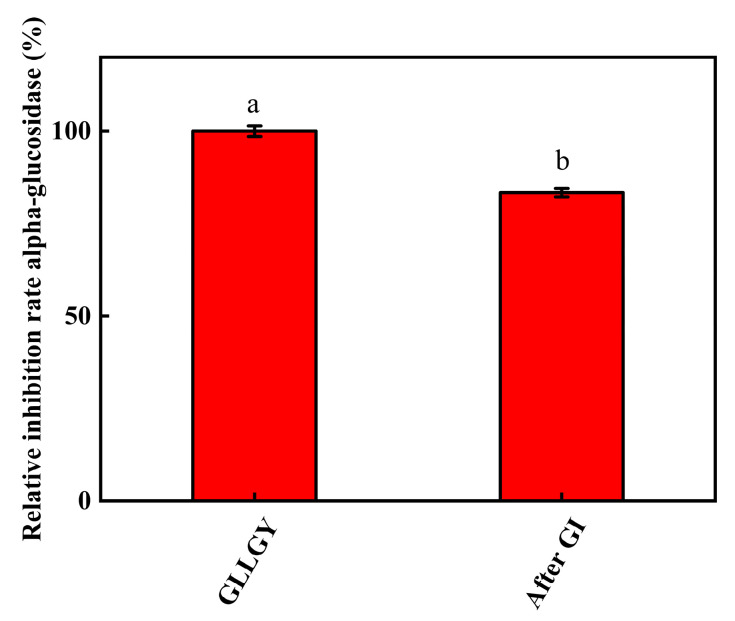
GLLGY α-glucosidase inhibition activity in a simulated gastrointestinal digestion system.

**Table 1 foods-12-00183-t001:** Identification of peptides in rice bran fermentation products by LC-MS/MS and molecular docking analysis of their potential binding sites with α-glucosidase.

Peptides	Scan Time (min)	Accurate Mass (Da)	Protein Family	Score	Intensity	Binding Energy (kcal/mol)	Binding Residues of α-Glucosidase	Hydrogen Bond Distance (Å)	Number of Hydrogen Bonds
Acarbose						−7.4	Asp282 Asp281 Asp404 Arg600 Asp616 His674	2.3/1.9 3.4/2.9 2.8 3.4/3.0 2.0/1.7 3.2/3.4	11
LFSGF	60.4865	570.292	ORYSJ Cellulose synthase-like protein	79.7	0	−8.0	Asp282 Ala284 Asp616	3.5 3.0 2.1/2.4/2.6/3.2	6
HWP	54.567	439.209	ORYSJ Mixed-linked glucan synthase	193.6	193.6	−7.8	Arg600 Asp616 His674	3.1 1.9/2.1/3.2 3.2	5
QSFF	55.3714	528.248	ORYSJ protein	286.6	336,170	−7.7	Asp282 Ser523 Asn524 Asp616	2.3/3.0/3.5 3.0 2.3/3.5 2.6	7
FPF	61.4677	410.207	BACSU PTS system oligo-beta-mannoside-specific EIIC component	85.3	0	−7.6	Asp616	1.9/3.4	2
FSGF	51.4358	457.209	ORYSJ Cellulose synthase-like protein	249.9	534,630	−7.6	Asp282 Asp284 Ser523 Asn524 Ala555	2.9 3.5 2.9/2.4/3.1 3.3 3.3	7
VYGF	50.7739	485.241	ORYSJ Cellulose synthase-like protein	219.3	982,140	−7.6	Asp282 Asp518 Asp616	2.4/3.5 2.5 2.3	4
GLIGY	50.3365	522.292	BACIU Phage tail length tape-measure protein	171.7	594,040	−7.2	Asp282 Asp404 Ser523 Ala555 Asp616 His674	2.0/2.9 2.1 1.8/3.3 2.5 1.9/3.3 2.9	9
GLLGY	50.3365	522.292	ORYSJ Phospholipase A1-II	171.7	594,040	−7.1	Asp282 Ser523 Asp616 His674	3.1 2.3 2.2/3.2 2.9	5
KGDPY	51.7567	579.293	BACSU Aspartokinase 1	88.8	0	−6.4	Asp282 Asp404 Asp616 His674	2.1/2.2 2.0 2.1/3.4 2.9	6
QSFLQRYYFLFRILPL	55.3714	2104.154	ORYSJ Protein kinase domain-containing protein	142.1	336,170	−6.4	Arg281 Asp282 Ala284 Pro285 Phe525 Arg527 Trp618 Glu622 Gln623	3.4/3.0/3.1 2.3/2.4 3.5 3.2 3.4 3.4 3.4 2.3 3.0	12
PSR	59.1375	359.218	ORYSA DNA-directed RNA polymerase subunit α	8.0	2,1243,000	−6.3	Asp282 Ser523 Arg600 Asp616	2.4/3.1/3.4/3.6 2.1/2.2/2.3 3.2 3.3/3.4	10
ISIFLSFFFGLIAGT	60.6946	1632.906	BACIU Amino acid ABC transporter permease	82.8	2,322,000	−6.2	Arg281 Asp282 Asp518 Asn524 Arg600	2.8/3.3 1.9 2.4/3.2 3.3/3.3 3.2	8

## Data Availability

The interaction data used to support the findings of this study are included within the article and the supporting information file. Also, all the data used to support the findings of this study are available from the corresponding author upon request.

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
