# Peer review of "Isolation of a Novel Anti-Diabetic α-Glucosidase Oligo-Peptide Inhibitor from Fermented Rice Bran"

_foods, 2023, doi:10.3390/foods12010183_

Round 1
Reviewer 1 Report
The paper "Isolation of a novel anti-diabetic α-glucosidase oligo-peptide inhibitor from fermented rice bran" submitted for review, addresses a medically important problem, the regulation of alpha- amylase activity and thus may help develop effective methods of blood glucose regulation.The Authors describe the subsequent steps of isolation and determination of the inhibitory potential of peptides obtained from rice bran after fermentiation with B. subtilis cells. I read the work with interest although I noticed a few shortcomings:
1. the nomenclature of enzymes should be standardized (alpha- or α-)
2. Similarly, when marking literature references-marking directly next to the text or after the space
3. centrifuged rather than rotated
4. The methods used should be described more clearly
5. Page 10- ...substrate- alterning enzyme conformation" not "confirmation"
6. In my opinion, the work should be checked by a native speaker
Author Response
The paper "Isolation of a novel anti-diabetic α-glucosidase oligo-peptide inhibitor from fermented rice bran" submitted for review, addresses a medically important problem, the regulation of alpha- amylase activity and thus may help develop effective methods of blood glucose regulation.The Authors describe the subsequent steps of isolation and determination of the inhibitory potential of peptides obtained from rice bran after fermentiation with B. subtilis cells. I read the work with interest although I noticed a few shortcomings:
Point 1: the nomenclature of enzymes should be standardized (alpha- or α-)
Response: Thank you very much for the commends. It has been revised in the manuscript.
Point 2: Similarly, when marking literature references-marking directly next to the text or after the space
Response: Thank you very much for the commends. It has been revised in the manuscript
Point 3: centrifuged rather than rotated
Response: Thank you very much for the commends. an equal volume of 10% TCA was added and the mixture was rotated at 4 °C for 1 h with a magnetic stirrer. After centrifugation at 8000 rpm, the obtained supernatant was subjected to rotary evaporation until no flowing liquid.
Point 4: The methods used should be described more clearly
Response: Thank you very much for the commends. It has been revised in the manuscript.
Point 5: Page 10- ...substrate- alterning enzyme conformation" not "confirmation"
Response: Thank you very much for the commends. It has been revised in the manuscript.
Point 6: In my opinion, the work should be checked by a native speaker
Response: Thank you very much for the commends. The manuscript has been edited by LEXIS Academic Editing Service.

Reviewer 2 Report
In general, the results of this work are interesting. It brings valuable findings about the anti-diabetic effects of new oligo-peptides isolated from fermented rice bran and suggests them as promising agents within anti-diabetic therapy.
However, some parts of this work necessarily need to be re-worked to enhance the interpretation of the obtained results and overall improve the quality of the paper:
Abstract
The abstract does not contain any background related to the topic of research. Also, the purpose of the study needs to be stated. Please, check the instruction for authors and add these essentials to the abstract. The formulation „anti-diabetic inhibitor“ (line 24) looks weird or could even be misleading. It should be rewritten to „anti-diabetic agent“or „alpha-glucosidase inhibitor“ or reformulate the whole sentence.
Introduction
In the broad context of this topic, in the introduction should be mentioned that not only peptides are the types of natural product with the alpha-glucosidase inhibition potential, but also phenolic compounds, terpenoids and non-starch polysaccharides have these activities (see, e.g. https://doi.org/10.1016/j.cofs.2018.02.008). Plenty of different peptides have also been studied with anti-diabetic potential from other natural and food sources like egg white, silkworms, black bean, quinoa and other sources. Some of these reports should be cited to show that the anti-diabetic potential of natural compounds, including peptides, has been intensively studied recently.
Methods
In two chapters related to use methods, several insufficient or unclear procedure steps must be added or elucidated, specifically:
Chapter 2.2
This chapter needs to provide complete information about cultivation conditions (temperature, length of cultivation or other cultivation conditions) to obtain the seed culture. These conditions are only described for the preparation of the fermented broth. In this, or the following chapter, the initial volumes of the fermentation media and the obtained supernatant need to be stated. These are important for assessing some of the further parameters used for the extraction of fermentation products.
Chapter 2.3
The sentence „Next, the obtained supernatant was evaporated in a rotary evaporator at 45 °C to reduce the mixture volume involving pausing once every 10 min.“ is unclear. The authors should clearly explain why the evaporation was paused every 10 minutes and how many times this step was performed. In the context of this sentence, the information about the total evaporation time and the remaining volume of the supernatant (before mixing with 10% TCA) within the experiment is also missing.
Results
This study provides comprehensive results. However, there are minor formal issues to be corrected. Figures 1 and 2 have not the explanatory title connected to the own figures. Between the figures and explanatory titles is the text of the manuscript. Table 1 is quite disordered, authors should consider changing the table's orientation to landscape format. The authors should state the presence of the acarbose and the related parameters due to the use of this compound as the positive control. Also, the presence of peptides „KGDPY“ and „QSFLQRYYFLFRILPL“ in the table is supported nowhere in the text of the manuscript. Their identification (+ PSR peptide) could be mentioned in Chapter 3.2, together with other identified peptides.
Discussion
The first paragraph of the discussion is not too well organized. Authors tried to assemble their thoughts and some general, topic-related findings, but they did not discuss them with the findings reported in the scientific literature. The first and partially also the second paragraph contains a lot of mostly raw results of the study, which also are not discussed with other literature. It leads to the fact that some results of the study only repeat in discussion without any further context. On the other hand, the last paragraph of the discussion is cogent and corresponds to the form of discussion. Specific commentaries and suggestions related to the discussion are point-by-point described below:
- „Many reports suggest that alpha-glucosidase inhibitory peptides exist in plants, animals, and microorganisms. Rice bran is rich in proteins[5].“ – This information is unnecessary in discussion when it is not further developed or discussed.
- „This study shows that bioactive peptides extracted from Bacillus subtilis fermented broth of rice bran can inhibit alpha-glucosidase activity (Fig. 1a).“ – What is the point of this statement in the discussion if it is not discussed with the scientific literature focused on anti-diabetic peptides derived from similar fermented products.
- „Bioactive peptides mainly inhibit the competition between oligosaccharides and alpha-glucosidase activity in small intestinal parietal cells, which slows down the intestinal absorption of glucose reducing blood glucose levels [33].“ – Also, this statement without further context fits more for the introduction.
- „After a series of purifications, we found that the bioactive peptides were mainly 370 <3 kDa. Thus, directly purifying components <3 kDa can save a lot of time.“ – It should be highlighted that this approach is preferably appliable to the bioactive / anti-diabetic peptides derived by fermentation from rice bran because it must not represent the most effective approach for the purification of other peptides from different natural sources.
- It is not explained in the text what the abbreviations „GSR“ (line 378) and „EAK“ (line 379) mean.
- Obtained data and alpha-glucosidase inhibition potential of oligo-peptides should be discussed with the other scientific literature (if exist) focused on their activities within in vivo models to support the potential application of oligo-peptides in the treatments for diabetes.
- In the discussion or the conclusions, some authors' opinions could be briefly suggested about future research and the potential issues related to the potential or broader use of anti-diabetic peptides (of rice bran or in general) as pharmaceuticals.
→ Based on the mentioned issue, I strongly recommend combining discussion with results to eliminate some repetitive parts related to (raw) results and the statements lacking deeper context. This change would also help the readers understand the particular results obtained within this research, reported already in the results chapter, in the deeper contexts and eliminate problematic continuity of the provided information.
Conclusion
The crucial results of this study are sufficiently sum-upped in the conclusions.
Author Response
In general, the results of this work are interesting. It brings valuable findings about the anti-diabetic effects of new oligo-peptides isolated from fermented rice bran and suggests them as promising agents within anti-diabetic therapy.
However, some parts of this work necessarily need to be re-worked to enhance the interpretation of the obtained results and overall improve the quality of the paper:
Point 1: Abstract
The abstract does not contain any background related to the topic of research. Also, the purpose of the study needs to be stated. Please, check the instruction for authors and add these essentials to the abstract. The formulation „anti-diabetic inhibitor“ (line 24) looks weird or could even be misleading. It should be rewritten to „anti-diabetic agent“or „alpha-glucosidase inhibitor“ or reformulate the whole sentence.
Response: Thank you very much for the commends. We have added background and revised in the abstract. At present, the incidence rate of diabetes is increasing gradually, and inhibiting α-glucosidase is one of the effective methods to control blood sugar. This study identified new peptides from rice bran fermentation broth and evaluated their inhibitory activity and mechanism against α-glucosidase. Rice bran was fermented with Bacillus subtilis MK15 and the polypeptides of <3 kDa were isolated by ultrafiltration and chromatographic column, which were then subjected to LC-MS/MS mass spectrometry analysis. The results revealed that the oligopeptide GLLGY showed the best inhibitory activity in vitro. Docking studies with GLLGY on human α-glucosidase (PDB ID 5NN8) suggested binding energy of -7.1 kcal/mol. GLLGY acts as a non-competitive inhibitor and forms 5 hydrogen bonds with Asp282, Ser523, Asp616, and His674 of α-glucosidase. Moreover, the peptide retained its inhibitory activity even in a simulated digestion environment in vitro. The oligo-peptide GLLGY could be developed into a potential anti-diabetic agent.
Point 2: Introduction
In the broad context of this topic, in the introduction should be mentioned that not only peptides are the types of natural product with the alpha-glucosidase inhibition potential, but also phenolic compounds, terpenoids and non-starch polysaccharides have these activities (see, e.g. https://doi.org/10.1016/j.cofs.2018.02.008). Plenty of different peptides have also been studied with anti-diabetic potential from other natural and food sources like egg white, silkworms, black bean, quinoa and other sources. Some of these reports should be cited to show that the anti-diabetic potential of natural compounds, including peptides, has been intensively studied recently.
Response: Thank you very much for the commends. We have revised in the introduction of manuscript. Diabetes is a complex multifactorial disease. Type 2 diabetes mellitus (T2DM) has become a serious health concern in developing countries [1], which is caused by insulin resistance due to altered β-insulin secretion or deficiency [2]. Worldwide, the incidence rate of diabetes is increasing rapidly, including in China. The development of diabetes is a very long process, which is often accompanied by other complications, such as diabetic foot, diabetic nephropathy, and cerebral thrombosis[2,3]. Delaying glucose absorption by inhibiting the activity of carbohydrate digestive enzymes such asα-glucosidase is one of the effective ways to control blood glucose levels [4]. So far, acarbose, voglibose, and miglitol are the three main α-glucosidase inhibitors used in the clinic[4]. These show good inhibitory effects but have certain side effects. Also, their long synthetic pathway makes it necessary to find otherα-glucosidase inhibitors with acceptable side effects. Natural products are rich in peptides, polysaccharides, Phenolic com-pounds, terpenoids and other bioactive substances, which can have a variety of functional properties[5,6]. The bioactive peptides extracted from Spirulina platensis[7], oat protein[8], canary seeds[9], quinoa[10] and other natural substances have inhibitory effects on α-glucosidase. In recent years, bioactive peptides among bioactive com-pounds have become the most important compounds for health and food industry, and can be used as substitutes for drug therapy to control lifestyle related diseases[11]. Therefore, the inhibition of α-glucosidase by bioactive peptides has a research prospect.
Point 3: Methods
In two chapters related to use methods, several insufficient or unclear procedure steps must be added or elucidated, specifically:
Chapter 2.2
This chapter needs to provide complete information about cultivation conditions (temperature, length of cultivation or other cultivation conditions) to obtain the seed culture. These conditions are only described for the preparation of the fermented broth. In this, or the following chapter, the initial volumes of the fermentation media and the obtained supernatant need to be stated. These are important for assessing some of the further parameters used for the extraction of fermentation products.
Response: Thank you very much for the commends. We have revised in the method of manuscript. B. subtilis MK15 was inoculated into a 100 mL seed medium (NA: beef extract 0.3%, peptone 1%, NaCl 0.5%, pH 7.0; sterilized at 121°C for 20 min), and then incubate at 37°C, 180 rpm and for 12 h. [24]. The 60 mL of seed culture was inoculated into 2 L of rice bran fermentation medium (rice bran 5%, glucose 0.1%, NaCl 0.5%, urea 0.5%, pH 8.0, sterilized at 115°C for 20 min), and then incubate at 35°C, 180 rpm for 48 h. The fermented broth was centrifuged at 8000 ×g and 4°C for 30 min to collect the supernatant [25]. The initial volume of fer-mentation medium was 2 L, and the supernatant obtained was 1.82 L.
Point 4: Chapter 2.3
The sentence „Next, the obtained supernatant was evaporated in a rotary evaporator at 45 °C to reduce the mixture volume involving pausing once every 10 min.“ is unclear. The authors should clearly explain why the evaporation was paused every 10 minutes and how many times this step was performed. In the context of this sentence, the information about the total evaporation time and the remaining volume of the supernatant (before mixing with 10% TCA) within the experiment is also missing.
Response: Thank you very much for the commends. We have revised in the method of manuscript. The supernatant was added with four times of absolute ethanol for precipitation at 4°C for 24 h and the mixture was centrifuged at 8000 rpm for 30 min. Next, the obtained supernatant was evaporated in a rotary evaporator at 45 °C to reduce the mixture volume, paused every 10 min (to prevent protein denaturation) and repeated about 10 times. 1.82 L supernatant, evaporation time was 2 h, and the remaining volume of the final supernatant was 150 mL. Subsequently, an equal volume of 10% TCA was added and the mixture was rotated at 4 °C for 1 h with a magnetic stirrer. After centrifugation at 8000 rpm, the obtained supernatant was subjected to rotary evaporation until no flowing liquid. Next, 50 mL of water was added for redissolution. Finally, after centrifugation, the obtained precipitate was freeze-dried (FreeZon®2.5L, Labconco, Kansas City, Missouri, USA)[26].
Point 5: Results
This study provides comprehensive results. However, there are minor formal issues to be corrected. Figures 1 and 2 have not the explanatory title connected to the own figures. Between the figures and explanatory titles is the text of the manuscript. Table 1 is quite disordered, authors should consider changing the table's orientation to landscape format. The authors should state the presence of the acarbose and the related parameters due to the use of this compound as the positive control. Also, the presence of peptides „KGDPY“ and „QSFLQRYYFLFRILPL“ in the table is supported nowhere in the text of the manuscript. Their identification (+ PSR peptide) could be mentioned in Chapter 3.2, together with other identified peptides.
Response: Thank you very much for the commends. It has been revised in the manuscript.
Point 6: Discussion
The first paragraph of the discussion is not too well organized. Authors tried to assemble their thoughts and some general, topic-related findings, but they did not discuss them with the findings reported in the scientific literature. The first and partially also the second paragraph contains a lot of mostly raw results of the study, which also are not discussed with other literature. It leads to the fact that some results of the study only repeat in discussion without any further context. On the other hand, the last paragraph of the discussion is cogent and corresponds to the form of discussion. Specific commentaries and suggestions related to the discussion are point-by-point described below:
- „Many reports suggest that alpha-glucosidase inhibitory peptides exist in plants, animals, and microorganisms. Rice bran is rich in proteins[5].“ – This information is unnecessary in discussion when it is not further developed or discussed.
- Response: Thank you very much for the commends. It has been revised in the manuscript.
- „This study shows that bioactive peptides extracted from Bacillus subtilis fermented broth of rice bran can inhibit alpha-glucosidase activity (Fig. 1a).“ – What is the point of this statement in the discussion if it is not discussed with the scientific literature focused on anti-diabetic peptides derived from similar fermented products.
- Response: Thank you very much for the commends. It has been revised in the manuscript.
- „Bioactive peptides mainly inhibit the competition between oligosaccharides and alpha-glucosidase activity in small intestinal parietal cells, which slows down the intestinal absorption of glucose reducing blood glucose levels [33].“ – Also, this statement without further context fits more for the introduction.
- Response: Thank you very much for the commends. It has been revised in the manuscript.
- „After a series of purifications, we found that the bioactive peptides were mainly 370 <3 kDa. Thus, directly purifying components <3 kDa can save a lot of time.“ – It should be highlighted that this approach is preferably appliable to the bioactive / anti-diabetic peptides derived by fermentation from rice bran because it must not represent the most effective approach for the purification of other peptides from different natural sources.
- Response: Thank you very much for the commends. It has been revised in the manuscript.
- It is not explained in the text what the abbreviations „GSR“ (line 378) and „EAK“ (line 379) mean.
- Response: Thank you very much for the commends. It has been revised in the manuscript.
- Obtained data and alpha-glucosidase inhibition potential of oligo-peptides should be discussed with the other scientific literature (if exist) focused on their activities within in vivo models to support the potential application of oligo-peptides in the treatments for diabetes.
- Response: Thank you very much for the commends. The α-glucosidase inhibitory peptides are attracted attention by food science researchers recently years. The determining of inhibitory effect still be studied in vitro. The investigation of inhibitory effect in vivo is the further required.
- In the discussion or the conclusions, some authors' opinions could be briefly suggested about future research and the potential issues related to the potential or broader use of anti-diabetic peptides (of rice bran or in general) as pharmaceuticals.
- Response: Thank you very much for the commends. It has been revised in the manuscript.
Based on the mentioned issue, I strongly recommend combining discussion with results to eliminate some repetitive parts related to (raw) results and the statements lacking deeper context. This change would also help the readers understand the particular results obtained within this research, reported already in the results chapter, in the deeper contexts and eliminate problematic continuity of the provided information.
Response: Thank you very much for the commends. we have revised in the manuscript. The extraction of active substances by microbial fermentation is emerging as a promising method [36]. Natural antidiabetic drugs had been extracted from microbial fermented food substances[37]. Many reports suggest that α-glucosidase inhibitory peptides exist in plants, animals, and microorganisms. Rice bran is rich in proteins[5]. Like the α-glucosidase inhibitor peptides obtained from fermented rubing cheese[38], chinese pickled chili pepper[39] and soybean condiment[40], the bioactive peptides extracted from rice bran fermentation broth can inhibit α-glucosidase. This study shows that bioactive peptides extracted from Bacillus subtilis fermented broth of rice bran can inhibit α-glucosidase activity (Fig. 1a). Bioactive peptides mainly inhibit the competition between oligosaccharides and α-glucosidase activity in small intestinal parietal cells, which slows down the intestinal absorption of glucose reducing blood glucose levels [41]. The short-chain peptides not only maintain their activity during gastrointestinal digestion and hydrolysis, but they are also suitable for design and chemical synthesis [33]. Several studies showed that peptides resistant to GI digestion have lower hydrophobicity, and more positive net charge at pH 7.0. [31] [42]. The average chain length and molecular weight of stable peptides are 4.5 ± 2.0 amino acid residues and 547.78 ± 233.17 g/mol, respectively, with a slightly positive net charge. Peptides with lower molecular weight may possess fewer protease recognition and cleavage sites [42]. The oligopeptide GLLGY is consistent with the peptide stability requirements of the in vitro GI digestion system. Moreover, the oligopeptide GLLGY maintained a good inhibitory effect on α-glucosidase in the simulated GI digestion system in vitro.
Point 7: Conclusion
The crucial results of this study are sufficiently sum-upped in the conclusions.
Response: Thank you very much for the commends. Rice bran was fermented by Bacillus subtilis MK15 to extract the peptide components. The rice bran peptides were identified by LC-MS/MS. Oligopeptide GLLGY showed the best inhibitory effect against α-glucosidase. Molecular docking results suggested that it forms 5 hydrogen bonds and van der Waals interactions involving the second Leu (GLLGY) with α-glucosidase, and GLLGY is a non-competitive inhibitor of α-glucosidase. In addition, GLLGY maintained an excellent α-glucosidase inhibitory effect in a GI digestion system. GLLGY can be a great anti-diabetic candidate and must be explored further. So far, the inhibitory effect only was studied in vitro; the effect in vivo still needs further exploration.

Round 2
Reviewer 2 Report
The manuscript has been revised adequately to the most of my commentaries and suggestions thus I recommend it for the publication in Foods.
Author Response
Point 1: The manuscript has been revised adequately to the most of my commentaries and suggestions thus I recommend it for the publication in Foods.
Response: Thank you very much for the commends.
